# Middle East Respiratory Syndrome Coronavirus (MERS-CoV) Seropositive Camel Handlers in Kenya

**DOI:** 10.3390/v12040396

**Published:** 2020-04-03

**Authors:** Alice N. Kiyong’a, Elizabeth A. J. Cook, Nisreen M. A. Okba, Velma Kivali, Chantal Reusken, Bart L. Haagmans, Eric M. Fèvre

**Affiliations:** 1International Livestock Research Institute, Old Naivasha Road, PO Box 30709, Nairobi 00100, Kenyae.cook@cgiar.org (E.A.J.C.); V.Kivali@cgiar.org (V.K.); 2Institute of Infection and Global Health, University of Liverpool, Leahurst Campus, Chester High Road, Neston CH64 7TE, UK; 3Viroscience Department, Erasmus Medical Center, 3015 GD Rotterdam, The Netherlands; n.okba@erasmusmc.nl (N.M.A.O.); C.Reusken@erasmusmc.nl (C.R.); b.haagmans@erasmusmc.nl (B.L.H.); 4Centre for Infectious Disease Control, National Institute for Public Health and the Environment, 3720 BA Bilthoven, The Netherlands

**Keywords:** coronavirus, camels, Republic of Kenya, slaughterhouses

## Abstract

Middle East respiratory syndrome (MERS) is a respiratory disease caused by a zoonotic coronavirus (MERS-CoV). Camel handlers, including slaughterhouse workers and herders, are at risk of acquiring MERS-CoV infections. However, there is limited evidence of infections among camel handlers in Africa. The purpose of this study was to determine the presence of antibodies to MERS-CoV in high-risk groups in Kenya. Sera collected from 93 camel handlers, 58 slaughterhouse workers and 35 camel herders, were screened for MERS-CoV antibodies using ELISA and PRNT. We found four seropositive slaughterhouse workers by PRNT. Risk factors amongst the slaughterhouse workers included being the slaughterman (the person who cuts the throat of the camel) and drinking camel blood. Further research is required to understand the epidemiology of MERS-CoV in Africa in relation to occupational risk, with a need for additional studies on the transmission of MERS-CoV from dromedary camels to humans, seroprevalence and associated risk factors.

## 1. Introduction

Middle East respiratory syndrome (MERS) is caused by an emerging beta-coronavirus (MERS-CoV). It is a zoonotic respiratory disease that was first reported in the Kingdom of Saudi Arabia in 2012 [1]. Dromedary camels are the reservoir of MERS-CoV [2] and contact with camels and their products is considered to be a risk factor for human MERS-CoV infection [3]. Occupational exposure has been reported in camel handlers, including camel farm workers and camel slaughterhouse workers [4], in the Middle East. It is hypothesized that the disease is transmitted from camels to people and person-to-person via respiratory secretions [5]. 

Previous research in Kenya has demonstrated a high MERS-CoV seropositivity in camels [6]. However, so far MERS-CoV neutralizing antibodies have only been detected in two non-camel-livestock keepers in Kenya [7]. In a targeted surveillance, none of the camel herders exposed to seropositive camels had MERS-CoV neutralizing antibodies [8].

In Kenya, populations living in semi-arid to arid environments have adopted camel rearing for drought resilience. Livestock keepers have a close relationship with their animals and cultural, animal husbandry and consumption practices may expose them to zoonotic agents. Poor hygienic conditions at farms and slaughterhouses, a lack of adequate infrastructure, the slaughtering and consumption of sick animals and weak monitoring and surveillance systems may contribute to an increased risk of exposure to MERS-CoV. There are limited data on MERS-CoV human infections in relation to occupational risk in Africa, with a need for additional studies on the transmission of MERS-CoV from dromedary camels to humans, seroprevalence and associated risk factors. The aims of this study were to determine the presence of antibodies to MERS-CoV in people in contact with camels and identify the risk factors associated with seropositivity.

## 2. Materials and Methods

The study was conducted in Isiolo, Laikipia and Machakos counties, Kenya, from October to November 2016 (Appendix A). Slaughterhouse workers and camel herders were recruited in Isiolo and Laikipia and slaughterhouse workers only in Machakos. These areas represent rural arid and semi-arid regions in Kenya where camels are kept and/or where camel slaughterhouses are located. Isiolo and Laikipia counties are in the arid and semi-arid northern region of Kenya and are inhabited by pastoral communities who keep camels as part of their livelihoods. Camels are transported to slaughterhouses in Isiolo, Laikipia and Machakos counties. 

The study was conducted in accordance with the Declaration of Helsinki, and the protocol was approved on August 17th 2016 by the Institutional Research Ethics Committee of the International Livestock Research Institute (IREC 2016-07), a committee approved by the Kenya National Commission for Science, Technology and Innovation. Permission to work in the slaughterhouses was granted by the Directorate of Veterinary Services for the Ministry of Agriculture, Livestock Development and Fisheries in Kenya. Written informed consent was sought from all the participants who agreed to take part in the study and anonymity and confidentiality were adhered to by using randomly assigned barcodes to label samples.

Data were collected from participants on their demographics and occupational and consumption practices using structured questionnaires from October to November 2016. Trained personnel collected clotted blood samples in 10 ml vacutainer tubes (Becton–Dickinson). The serum was separated by centrifugation at 1150× *g* for 20 minutes using a Beckman Coulter Avanti J-E centrifuge. The sera samples were stored in duplicate at −20 °C until testing. 

All the sera were tested for the presence of MERS-CoV specific antibodies using the commercial Euroimmun Anti-MERS-CoV ELISA at the International Livestock Research Institute (ILRI) laboratory in Nairobi. The positive and negative controls were human sera supplied by the manufacturer. The tests were carried out as per the manufacturers’ instructions. The extinction value or optical density of each analyzed sample was divided by the extinction value of the calibrator (supplied by the manufacturer) to calculate an extinction ratio. Samples with an extinction ratio of 0.3 were considered positive as previously described [7]. 

The sera were tested for the presence of MERS-CoV spike specific antibodies using an in-house S1 ELISA at the Erasmus MC Viroscience Laboratory in Rotterdam according to the previously validated protocol [9].

Twenty-one samples, including all Euroimmun and in-house S1 ELISA positive samples and a random selection of negative samples, were tested for the presence of MERS-CoV neutralizing antibodies using PRNT as described earlier with some modifications [9]. The positive and negative controls were the same as those described previously [9]. The testing was performed at the Erasmus MC Viroscience Laboratory in Rotterdam. Heat-inactivated sera were serially diluted in an RPMI1640 medium supplemented with penicillin, streptomycin and 1% fetal bovine serum (starting 1:10), mixed 1:1 with MERS-CoV (EMC/2012; 400 PFU) and incubated at 37 °C for one hour. Following that, the mix was transferred to a monolayer of HuH-7 cells in 96-well plates and incubated at 37 °C for one hour. The mix was then removed and the cells were further incubated at 37 °C for eight hours. The cells were then fixed and stained using an anti-MERS-CoV N protein mouse monoclonal antibody (1:5000, Sino Biological) and a secondary peroxidase-labelled goat anti-mouse IgG1 (1:2000, SouthernBiotech). The signal was developed using a precipitate forming TMB substrate (True Blue, KPL). The numbers of infected cells per well were counted using the ImmunoSpot® Image analyzer (CTL Europe GmbH). The neutralization titer of each serum sample was determined as the reciprocal of the highest dilution resulting in a ≥50% reduction in the number of infected cells (PRNT50). A titer of ≥20 was considered to be positive.

The statistical analysis was performed in R (http://CRAN.R-project.org/). To identify the risk factors associated with MERS-CoV seropositivity, a univariable analysis was conducted using Pearson’s chi-square with a Monte Carlo simulation of 10,000 repetitions to account for the small sample size. The results of the PRNT were used as a final determination of sero-reactivity. The statistical significance was set at *p* = 0.05 and confidence intervals were determined using a standard error of 1.96.

## 3. Results

A total of 58 slaughterhouse workers were sampled from three slaughterhouses; 5 were from Machakos, 16 from Laikipia and 37 from Isiolo (Appendix A). The majority of workers were male (*n* = 47) and the mean age was 37 years (range 17–73 years). In Machakos, 8–10 camels were slaughtered daily, in Laikipia 5 camels were slaughtered one day per week and in Isiolo 10–20 camels were slaughtered two days per week. 

Thirty-five camel herders were sampled—33 in Isiolo and 2 in Laikipia. The majority of herders were male (*n* = 30) and the mean age was 45 years (range 2–82 years). The mean number of camels owned was 36 (range 0–149). The ratio of juvenile camels (less than 12 months) to adults in herds was 1:1.4. 

The results of the three serological tests are presented in Table 1. Samples from five slaughterhouse workers (8.6%; 95% CI 3.8%–19.0%) and three camel herders (8.6%; 95% CI 8.6%–22.1%) were seropositive for MERS-CoV when tested using the Euroimmun ELISA. The samples were retested with the S1 ELISA and four slaughterhouse workers (6.9%; 95% CI 2.8%–16.3%) demonstrated antibodies to MERS-CoV compared to one camel herder (2.9%; 95% CI 0.7%–14.2%). 

Four of the twenty-one samples tested using PRNT were positive. The agreement between the tests is shown in Appendix A and Figure 1. All the PRNT positive samples originated from slaughterhouse workers in Isiolo, where the apparent prevalence was 10.8% (95% CI 4.4%–24.6%). 

The proportion of PRNT positive samples at the Isiolo slaughterhouse is demonstrated in Appendix A. A risk factor analysis was conducted on samples from the Isiolo slaughterhouse only. Only PRNT positive samples were considered in the analysis. The proportion of positive male slaughterhouse workers (3/26) was not significantly different to the proportion of positive female workers (1/11) (Chi^2^ = 0.04, *p* = 1). There was a higher but not significant proportion of positive workers aged 40 years and over (2/12) compared to those less than 40 years (2/25) (Chi^2^ = 0.49, *p* = 0.60). The three positive men were slaughtermen, meaning they were responsible for the slaughtering event, and the positive female worker had another role in the slaughterhouse. The proportion of positive slaughtermen (3/8) was significantly different to that of other roles in the slaughterhouse (1/29) (Chi^2^ = 5.2, *p* = 0.05). Drinking camel blood was also significantly associated with seropositivity (3/6) compared to those who did not drink camel blood (1/31) (Chi^2^ = 7.3, *p* = 0.03). Other non-significant exposures included drinking camel milk (3/22) compared to not consuming the milk (1/15) (Chi^2^ = 0.37, *p* = 0.64) and milking camels (3/14) compared to not milking camels (1/23) (Chi^2^ = 2.05, *p* = 0.28).

## 4. Discussion

This is the first report of MERS-CoV neutralizing antibodies in camel handlers in Kenya. We detected an apparent prevalence of 10.8% MERS-CoV seropositivity by PRNT in slaughterhouse workers working in a camel-keeping area of the country. The detection of MERS-CoV in dromedary camels in Kenya in recent years [10] has highlighted the potential for transmission of the virus to people in close contact with camels as reported in the Middle East [4], but previous research investigating high-risk groups failed to detect individuals who were seropositive by virus neutralization [8]. The clinical significance of detecting MERS-CoV seropositivity by PRNT in our population is unknown. This was a cross-sectional serosurvey of healthy workers; we highlight that these individuals did not show clinical signs of disease and we are unable to determine when the exposures took place. The low virus neutralizing antibody titers may suggest asymptomatic infections [4], and clinical infections, which most likely present as transient infections, may be misdiagnosed as other endemic diseases [11]. 

Studies investigating the potential for transmission of MERS-CoV from camels to high-risk groups in other regions have had variable findings. Despite virus detection in camels at slaughterhouses in Nigeria, neutralizing antibodies were not detected in slaughterhouse workers [12]. Similarly, neutralizing antibodies have not been detected in slaughterhouse workers in the Kingdom of Saudi Arabia [13]. In contrast, virus neutralizing antibodies were detected in slaughterhouse workers in Qatar, where active MERS-CoV shedding has been demonstrated in slaughter animals [4]. A substantial pool of susceptible animals is necessary to support virus transmission and result in a risk to people [2]. Susceptible animals brought together for slaughter from different regions may drive virus amplification and zoonotic transmission [14]. 

Our previous research in Kenya has demonstrated a high seroprevalence of MERS-CoV antibodies in camels [6]. However, research in Kenya and elsewhere has demonstrated that juvenile camels have a higher rate of viral RNA positivity than adult animals [10,15]. In Kenya, the primary purpose of camel keeping is for milk production and therefore only mature animals (greater than three years) are presented for slaughter. This may limit the potential for transmission of the virus to slaughterhouse workers and explain the small number of positive samples in this study. Further studies targeting camel handlers who work with younger animals are required.

The currently available commercial ELISA is the Euroimmun Anti-MERS-CoV. This study demonstrates that even when using a low cut-off as recommended [7], the test is less specific and sensitive in detecting MERS-CoV seropositives compared to our in-house S1 ELISA, as previously observed [9]. However, there was a good correlation between the results of the in-house S1 ELISA and the PRNT. A sample reactive in both the S1 ELISA and PRNT was considered to be positive. Having a sensitive assay is crucial to avoid errors in the estimation of prevalence in seroepidemiological studies.

In this study, the small sample size makes it difficult to draw conclusions about risk factors for MERS-CoV seropositivity. The sample size is limited because the population of camel slaughterhouse workers is small. The statistical analysis accounted for the small sample size, but the results should be interpreted with caution. Potential risk factors that might be investigated in future studies include being a slaughterman (the person who cuts the animal’s throat). This has been reported for other zoonotic viruses transmitted by the respiratory route, including Rift Valley fever (RVF) [16].

Other risk factors that should be investigated include drinking camel blood [3]. This has not been significantly associated with MERS-CoV seropositivity but has been reported for RVF, another zoonotic RNA virus [16]. The milking of camels has also been reported by other studies as a risk factor for MERS-CoV and this needs further evaluation [3]. Further targeted studies investigating these and other risk factors in larger populations over longer periods of time are required.

## Figures and Tables

**Figure 1 viruses-12-00396-f001:**
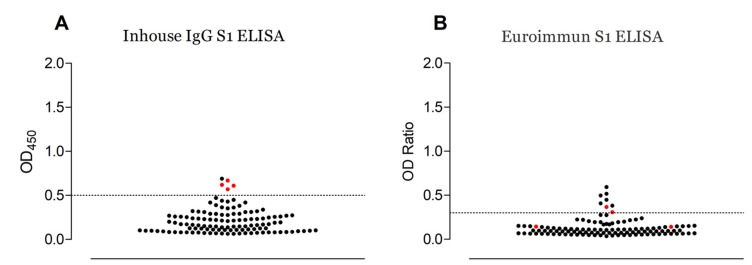
The testing of human serum samples for MERS-CoV antibodies, Kenya, (2016). The reactivities of individual serum samples to MERS-CoV S1—tested using in-house (**A**) and commercial (**B**) S1 ELISAs—are plotted. PRNT positive serum samples are shown in red. The dotted line indicates the cut-off of each assay [7,9].

**Table 1 viruses-12-00396-t001:** The proportion of slaughterhouse workers and camel herders who were positive for MERS-CoV antibodies when tested by the Euroimmun ELISA, S1 ELISA and PRNT.

Cohort	Number of Samples	Euroimmun S1 ELISA Positive Number (%)	In-house S1 ELISA Positive Number (%)	PRNT_50_ Positive Number	
				Commercial S1 ELISA positive	In-House S1 ELISA Positive
Slaughterhouse workers	58	5 (8.6)	4 (6.8)	2/5	4/4
Camel herders	35	3 (8.6)	1 (2.9)	0/3	0/1
Total	93	8 (8.6)	5 (5.4)	2/8	4/5

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
