# Peer review of "Middle East Respiratory Syndrome Coronavirus (MERS-CoV) Seropositive Camel Handlers in Kenya"

_viruses, 2020, doi:10.3390/v12040396_

Round 1
Reviewer 1 Report
The authors describe a serological survey to detect the frequency of exposure to Middle East respiratory syndrome coronavirus (MERS-CoV) among 93 people in close daily contact with camels in Kenya. The authors used three different serological tests to assess the MERS-antibody status of the sera. All 93 available sera were tested by Euroimmun Anti-MERS-CoV ELISA and in-house S1 ELISA. Samples positive in any of these two tests and a random selection of negative samples were re-tested using PRNT. Based on results presented in Supplementary Table 1, 10 samples were positive for MERS CoV antibody in at least one of the serological tests used, with only 2 samples tested positive in all three tests.
While the methodology appears sound, and the topic should be of interest to the readers of the journal, I have some concerns about interpretation of the data presented and conclusions reached by the authors:
- The agreement between all three tests appears to be poor, which is a little bit concerning. While some differences between testing results are to be expected due to different sensitivities/specificities of the tests used, it would add confidence to the data if the authors included a set of known positive and known negative samples in their sample pool tested with all three tests.
- Further to point 1 above, what was used as positive and negative controls for the tests? Did they show expected results?
- The authors should define what samples they considered positive in the final analysis – it appears that only samples positive by PRNT were considered positive, but this needs to be stated.
- With only 4 positive individuals, the value of the statistical analysis on page 109 is questionable.
- The main finding of the paper is a low seropositivity to MERS CoV among people who have daily contact with camels in Kenya. However, as the authors did not collect blood and/or nasal swabs from camels at the same time, it is difficult to draw any definite conclusions from these data. It may be that MERS-CoV is not easily transmitted from camels to people (which these data would suggest), or that the prevalence of active MERS-CoV infection among the relevant camel populations was low. Viral RNA was detected in nasal swabs from only 7/2175 (0.23%) camels from two herds in the Isiolo county based on data from the paper cited by the authors to support the presence of MERS-CoV in African camels (Kiambi et al. 2018). There are several other papers that showed high seroprevalence of MERS-CoV among dromedaries in various places of Africa, but regional differences seem to exist. It may be relevant to include a citation to a recent literature review of the topic (Dighe at al 2019, https://www.sciencedirect.com/science/article/pii/S1755436519300416)
- What was the timeframe for collection of samples (dates)?
- It would be relevant to include some data on the camel populations that the participants of the survey were exposed to e.g. age ranges of animals, the size of the herds or the number of camels slaughtered on a daily/weekly basis, the level of animal mixing/imports…etc, as those factors are likely to be relevant for assessing the likelihood of MERS-CoV circulation among the animals in the absence of the actual serological/virological data.
- Some minor editorial-type comments:
Line 15: “Middle East respiratory syndrome” instead of “Middle East Respiratory Syndrome”
Line 26: delete “viral” (redundant)
Line 38: “…may expose them to zoonotic agents” instead of “zoonotic diseases”
Line 39: “conditions at farms” not “in farms”
Line 40: “…and weak monitoring and surveillance systems may contribute to an increased risk of exposure to MERS-CoV” or something to that effect, instead of the current version (“may expose this group to MERS-CoV)
Lines 41-43: Please re-word this so that the aims of the study are clearly stated.
Line 62: Delete “and”
Line 64: Insert spaces after “3000” and “20”. Also, the type of the centrifuge needs to be specified when providing the speed in rpm. It is better to state the centrifugation speed in centrifugal force (“x g”), as this is independent of the centrifuge radius.
Line 69: What is “extinction ratio”?
Line 76-77: Delete “Sera were tested for the presence of MERS-CoV neutralzing antibodies” as this is redundant. The first two sentences could be re-worded for example to: “Twenty-one samples including…[…]…were tested for the presence of MERS-CoV neutralizing antibodies using PRNT as described earlier with some modifications [9]. The testing was performed a the Erasmus MC Viroscience Laboratory in Rotterdam”
Line 79: What were the sera diluted in?
Line 81: What were the conditions for cell incubations for 8 hours?
Lines 83-84: What were the dilutions of commercial antibody and what was used as a diluent?
Line 89: Delete “in” after “was”.
Line 93: My suggestion would be to re-write this sentence to “Samples from x workers and y camel herders were seropositive for MERS-CoV when tested by xx test”.
Line 99: “The proportion of slaughterhouse workers and camel herders that were positive for MERS-CoV antibodies by…” would read better than the current version.
Table 1: The total under PRNT50 is misleading. It should not be 2/93 or 4/93, as only 21 samples were tested using this test.
Line 110: “is shown” rather than “is demonstrated”. As stated above, the value of this entire paragraph is somewhat questionable with only 4 positive samples.
Table S1: The meaning of shaded fields needs to be stated e.g. “Positive results are shaded” or something to that effect.
Table S2: There needs to be consistency in the way results are presented in the table e.g. for columns Machakos and Laikipia the data are presented as single numbers (“x”), while for the column Isiolo the data are presented as “x/y (z)”
Author Response
Many thanks for the prompt review of our manuscript. We have addressed the reviewers’ comments which are outlined below.
The line numbers refer to the marked-up version of the manuscript.
On behalf of all authors.
Alice Kiyong’a
Reviewer 1.
- The agreement between all three tests appears to be poor, which is a little bit concerning. While some differences between testing results are to be expected due to different sensitivities/specificities of the tests used, it would add confidence to the data if the authors included a set of known positive and known negative samples in their sample pool tested with all three tests.
RESPONSE: Regarding the Euroimmun ELISA the positive and negative controls were human sera supplied by the manufacturer (line 101-102) and for the other tests the sera used are described in Okba et al 2019 (line 111-112). Details regarding this have been added to the methods section.
- Further to point 1 above, what was used as positive and negative controls for the tests? Did they show expected results?
RESPONSE: please see the point above
- The authors should define what samples they considered positive in the final analysis – it appears that only samples positive by PRNT were considered positive, but this needs to be stated.
RESPONSE: this information has been added to the methods line 130 and the discussion line 244-245.
- With only 4 positive individuals, the value of the statistical analysis on page 109 is questionable.
RESPONSE: The authors agree that due to the small samples size and the small number of positives that statistical analysis and interpretation are limited. The sample size is limited by the number of workers available. We have changed the analysis to include Monte Carlo simulation to account for the small cell sizes line 129 and added a comment to the discussion regarding the interpretation line 248-250.
- The main finding of the paper is a low seropositivity to MERS CoV among people who have daily contact with camels in Kenya. However, as the authors did not collect blood and/or nasal swabs from camels at the same time, it is difficult to draw any definite conclusions from these data. It may be that MERS-CoV is not easily transmitted from camels to people (which these data would suggest), or that the prevalence of active MERS-CoV infection among the relevant camel populations was low. Viral RNA was detected in nasal swabs from only 7/2175 (0.23%) camels from two herds in the Isiolo county based on data from the paper cited by the authors to support the presence of MERS-CoV in African camels (Kiambi et al. 2018). There are several other papers that showed high seroprevalence of MERS-CoV among dromedaries in various places of Africa, but regional differences seem to exist. It may be relevant to include a citation to a recent literature review of the topic (Dighe at al 2019, https://www.sciencedirect.com/science/article/pii/S1755436519300416)
RESPONSE: Thank you for these comments.
Our own study (Deem et al, 2015) shows that there is a high (50%+) seroprevalence of MERS-CoV antibodies in herds including animals at slaughter in the area where we were collecting the data for this paper.
However, it has been shown in previous studies that juvenile animals are more likely to be RNA positive. Additional comments have been added to the discussion explaining that animals at slaughter are likely to be older and may not be excreting virus, which may explain the small numbers of seropositive slaughterhouse workers detected line 233-239.
- What was the timeframe for collection of samples (dates)?
RESPONSE: this has been added to the methods line 69
- It would be relevant to include some data on the camel populations that the participants of the survey were exposed to e.g. age ranges of animals, the size of the herds or the number of camels slaughtered on a daily/weekly basis, the level of animal mixing/imports…etc, as those factors are likely to be relevant for assessing the likelihood of MERS-CoV circulation among the animals in the absence of the actual serological/virological data.
RESPONSE: As mentioned above our own study (Deem et al, 2015) shows that there is a high (50%+) seroprevalence of MERS-CoV antibodies in herds including animals at slaughter in the area where we were collecting the data for this paper, and hence the need to explore risks to humans.
Further details have been added to the results from information collected that covers the age and number of animals line 133-141. A comment has also been added to the discussion regarding the potential relationship between age of camels at slaughter and potential for viral RNA positivity line 233-239.
- Some minor editorial-type comments:
Line 15: “Middle East respiratory syndrome” instead of “Middle East Respiratory Syndrome”
RESPONSE: This has been done for abstract and introduction line 16 and line 37
Line 26: delete “viral” (redundant)
RESPONSE: This has been done line 38
Line 38: “…may expose them to zoonotic agents” instead of “zoonotic diseases”
RESPONSE: This has been done line 50
Line 39: “conditions at farms” not “in farms”
RESPONSE: This has been done line 51
Line 40: “…and weak monitoring and surveillance systems may contribute to an increased risk of exposure to MERS-CoV” or something to that effect, instead of the current version (“may expose this group to MERS-CoV)
RESPONSE: This has been done line 52
Lines 41-43: Please re-word this so that the aims of the study are clearly stated.
RESPONSE: A sentence clarifying the aims has been added line 56-57
Line 62: Delete “and”
RESPONSE This has been done line 86
Line 64: Insert spaces after “3000” and “20”. Also, the type of the centrifuge needs to be specified when providing the speed in rpm. It is better to state the centrifugation speed in centrifugal force (“x g”), as this is independent of the centrifuge radius.
RESPONSE: This has been changed to g and spaces added line 88-89
Line 69: What is “extinction ratio”?
RESPONSE: A sentence has been added to describe this line 102-104
Line 76-77: Delete “Sera were tested for the presence of MERS-CoV neutralzing antibodies” as this is redundant. The first two sentences could be re-worded for example to: “Twenty-one samples including…[…]…were tested for the presence of MERS-CoV neutralizing antibodies using PRNT as described earlier with some modifications [9]. The testing was performed a the Erasmus MC Viroscience Laboratory in Rotterdam”
RESPONSE: This has been done line 109-111
Line 79: What were the sera diluted in?
RESPONSE: This has been added to the methods line 114
Line 81: What were the conditions for cell incubations for 8 hours?
RESPONSE: This has been added to the methods line 115, 117 and 118
Lines 83-84: What were the dilutions of commercial antibody and what was used as a diluent?
RESPONSE: This has been added line 119-120
Line 89: Delete “in” after “was”.
RESPONSE: This has been done line 127
Line 93: My suggestion would be to re-write this sentence to “Samples from x workers and y camel herders were seropositive for MERS-CoV when tested by xx test”.
RESPONSE: This has been done line 142-144
Line 99: “The proportion of slaughterhouse workers and camel herders that were positive for MERS-CoV antibodies by…” would read better than the current version.
RESPONSE: This has been done line 148
Table 1: The total under PRNT50 is misleading. It should not be 2/93 or 4/93, as only 21 samples were tested using this test.
RESPONSE: This has been changed
Line 110: “is shown” rather than “is demonstrated”. As stated above, the value of this entire paragraph is somewhat questionable with only 4 positive samples.
RESPONSE: This has been done line 161. We modified the analysis and noted the limitations in the discussion line 129 and 248-250.
Table S1: The meaning of shaded fields needs to be stated e.g. “Positive results are shaded” or something to that effect.
RESPONSE: This has been added
Table S2: There needs to be consistency in the way results are presented in the table e.g. for columns Machakos and Laikipia the data are presented as single numbers (“x”), while for the column Isiolo the data are presented as “x/y (z)
RESPONSE: This has been split into two tables
Reviewer 2 Report
Kiyong'a et al. performed a study that evaluated the extent of seropositivity of camel herders and camel slaughterhouse workers to MERS-CoV, a deadly human coronavirus. There study utilized different approaches to quantify MERS-CoV antibodies present and collected personal information to help derive the potential sources of inoculation with MERS-CoV. This study is the first of its kind to look at seropositivity in these populations in Kenya. Major Comments: 1. Figure 1 seems to set an arbitrary cutoff for seropositivity. Yet, there does not appear to be a significant difference between titers deemed seropositive versus seronegative. Furthermore, the tests don't both conclusively demonstrate seropositivity for the 4 reported cases (in red). The authors do suggest that if these individuals are seropositive, that the lower titers were likely due to an asymptomatic case. However, it would be helpful to readers to explain what justification was used to establish the cutoff lines for seropositivity. With 4 positive cases, it is surprising that all were asymptomatic. Minor Comments: 1. Line 89: “Statistical analysis was performed in R.” There is an extra “in” in this sentence.Author Response
Many thanks for the prompt review of our manuscript. We have addressed the reviewer’s comments which are outlined below.
The line numbers refer to the marked-up version of the manuscript.
On behalf of all authors.
Alice Kiyong’a
Reviewer 2
Major Comments: 1. Figure 1 seems to set an arbitrary cutoff for seropositivity. Yet, there does not appear to be a significant difference between titers deemed seropositive versus seronegative. Furthermore, the tests don't both conclusively demonstrate seropositivity for the 4 reported cases (in red). The authors do suggest that if these individuals are seropositive, that the lower titers were likely due to an asymptomatic case. However, it would be helpful to readers to explain what justification was used to establish the cutoff lines for seropositivity.
RESPONSE; The cut off for the Euroimmun ELISA was described in Liljander at al 2016 (line 105) and the cut-off for the S1 ELISA in Okba et al 2019 (line 108). This is described in the methods and we have added the references to the figure description. There is good agreement between the S1 ELISA and the PRNT shown in Table 1 and highlighted in the Discussion (line 243).
With 4 positive cases, it is surprising that all were asymptomatic.
RESPONSE: This was a cross-sectional serosurvey of healthy workers, not a study of patients, so we would not have expected participants to be symptomatic. This has been noted in the discussion line 202-204.
Minor Comments: 1. Line 89: “Statistical analysis was performed in R.” There is an extra “in” in this sentence.
RESPONSE: This has been corrected line 127